# Lithium Chloride and GSK3 Inhibition Reduce Aquaporin-2 Expression in Primary Cultured Inner Medullary Collecting Duct Cells Due to Independent Mechanisms

**DOI:** 10.3390/cells9041060

**Published:** 2020-04-23

**Authors:** Marc Kaiser, Bayram Edemir

**Affiliations:** 1Medizinische Klinik D, Experimentelle Nephrologie, Universitätsklinikum Münster, 48143 Münster, Germany; m.jardzejewski@googlemail.com; 2Department of Medicine, Hematology and Oncology, Martin Luther University Halle-Wittenberg, 06120 Halle (Saale), Germany

**Keywords:** aquaporin, diabetes insipidus, lithium chloride

## Abstract

Lithium chloride (LiCl) is a widely used drug for the treatment of bipolar disorders, but as a side effect, 40% of the patients develop diabetes insipidus. LiCl affects the activity of the glycogen synthase kinase 3 (GSK3), and mice deficient for GSK3β showed a reduction in the urine concentration capability. The cellular and molecular mechanisms are not fully understood. We used primary cultured inner medullary collecting duct cells to analyze the underlying mechanisms. LiCl and the inhibitor of GSK3 (SB216763) induced a decrease in the aquaporin-2 (Aqp2) protein level. LiCl induced downregulation of Aqp2 mRNA expression while SB216763 had no effect and TWS119 led to increase in expression. The inhibition of the lysosomal activity with bafilomycin or chloroquine prevented both LiCl- and SB216763-mediated downregulation of Aqp2 protein expression. Bafilomycin and chloroquine induced the accumulation of Aqp2 in lysosomal structures, which was prevented in cells treated with dibutyryl cyclic adenosine monophosphate (dbcAMP), which led to phosphorylation and membrane localization of Aqp2. Downregulation of Aqp2 was also evident when LiCl was applied together with dbcAMP, and dbcAMP prevented the SB216763-induced downregulation. We showed that LiCl and SB216763 induce downregulation of Aqp2 via different mechanisms. While LiCl also affected the mRNA level, SB216763 induced lysosmal degradation. Specific GSK3β inhibition had an opposite effect, indicating a more complex regulatory mechanism.

## 1. Introduction

In the renal collecting duct, the final urine concentration is predominantly regulated by the action of the antidiuretic hormone vasopressin (AVP), which is secreted by the neuronal lobule of the pituitary gland [1]. The binding of AVP to the vasopressin type 2 receptor (V_2_R) at the basolateral membrane of the collecting duct’s principal cells leads to, via an activation of a G_s_-protein, the stimulation of the adenylyl cyclase and an increase of the cyclic adenosine monophosphate (cAMP) level [2]. This activates the protein kinase A (PKA), which in turn phosphorylates the water channel aquaporin-2 (Aqp2) at serine 256 [3]. This phosphorylation induces the translocation and fusion of Aqp2-containing vesicles with the luminal plasma membrane. Thus, water can enter the cell along an osmotic gradient and leaves the cell on the basolateral site facilitated by aquaporin-3 (Aqp3) and aquaporin-4 (Aqp4) [4]. Defects in this pathway often lead to a urine concentration deficiency, a disorder called diabetes insipidus (DI) [5]. This disorder can be inherited, either when specific mutations lead to a defect in the secretion of AVP from the pituitary gland (central DI) or to nephrogenic diabetes insipidus (NDI) when the V_2_R or Aqp2 genes are mutated [5]. Alternatively, DI can be acquired, e.g., due to therapy with the drug lithium chloride (LiCl) or due to diseases affecting the pituitary gland [6,7]. LiCl is widely used for the treatment of bipolar disorders and causes NDI in up to 40% of these patients as severe side effect [8]. Various mechanisms have been described of how LiCl could induce NDI. One postulated mechanism is the stimulation of the G_i_ protein by LiCl, which would inhibit the activity of the adenylate cyclase [8]. As a consequence, the trafficking of Aqp2-containing vesicles to the apical membrane and Aqp2 expression, which is also cAMP and protein kinase A (PKA) dependent, would be reduced [9]. Another postulated mechanism is that LiCl reduces the density of the V_2_R at the plasma membrane, which would have the same effect as described above. Moreover, LiCl might cause a defect independent of the classical V_2_R pathway [10,11]. LiCl, for example, inhibits the activity of the glycogen synthase kinase 3 alpha/beta (GSK3α/β), but it is still not well understood how the inhibition of GSK3α/β by LiCl causes NDI [12]. A global GSK3β-deficient mouse was embryonic lethal [13], and in contrast, a renal collecting duct specific knockout mouse showed only mild polyuria. However, under water deprivation or stimulation with AVP, reduced cAMP levels and in turn a urine concentration disability was observed [14]. Therefore, it was concluded that GSK3β is involved in the urine concentration under water-deprived conditions. Still, the underlying cellular and molecular mechanisms were not fully understood. To further address this issue, we used primary cultured inner medullary collecting duct (IMCD) cells and analyzed the effects of pharmacological GSK3 inhibition and LiCl on Aqp2 expression. The treatment of IMCD cells with LiCl provoked a decrease of Aqp2 expression on both the transcriptional and protein level. Inhibition of both isoforms of the GSK3 with SB216763, however, decreased only the expression of Aqp2 protein. In contrast, a sole inhibition of GSK3β by the specific inhibitor, TWS119 [15], showed an upregulation of Aqp2 on the protein level as well as in mRNA expression. Further analysis showed that LiCl and GSK3 inhibition by SB216763 enhanced the degradation rate of Aqp2 protein in lysosomes.

## 2. Materials and Methods

### 2.1. Cell Culture

Primary cultured IMCD cells were prepared as previously described [16]. Female Wistar rats at the age of 23 months were sacrificed by decapitation and the kidneys were removed. The inner medulla and the papilla were isolated, cut into small pieces, and incubated at 37 °C for 90 min in a digesting solution, which contained phosphate buffered saline (PBS) (Biochrom, Berlin, Germany), 0.2% hyaluronidase (Sigma, Deisenhofen, Germany), and 0.2% collagenase type CLS-II (Sigma). The isolated cells were seeded with a density of approximately 10^5^ cells/cm^2^ and grown to confluency in 24-well plates (Greiner Bio-One, Frickenhausen, Germany) coated with collagen type IV (Becton-Dickinson, Heidelberg, Germany). For cultivation, Dulbecco’s modified Eagle’s medium (DMEM) containing 100 IU/mL penicillin and 100 μg/mL streptomycin, 0.2% glutamine, 1% non-essential amino acids, and 1% ultroser (BioSepra Inc., Marlborough, MA, USA) was used. The medium osmolality was increased to 600 mOsmol/kg by the addition of 100 mM NaCl and 100 mM urea. Osmolality was checked using an osmometer (Knauer, Berlin, Germany). Cells were cultured for 67 days and treated with LiCl; SB216763, a potent inhibitor of both GSK3 isoforms [17] (Sigma, Deisenhofen, Germany); TWS119, a potent inhibitor of the GSK3β isoform [15] (Millipore, Darmstadt, Germany); bafilomycin A1 and chloroquine, both inhibitors of the vacuolar ATPase [18,19] and thereby lysosomal activity inhibitors; MG132, as a proteasome inhibitor [20]; and dbcAMP (all from Sigma, Deisenhofen, Germany) in different concentrations and durations, alone or together as indicated. All experiments were conducted with at least 3 independent cell cultures from different rats. All animals were handled according to German animal protection guidelines and the experiments were approved by a governmental committee on animal welfare (A 60/1993 and A 67/09).

### 2.2. Immunoblotting

Immunoblotting was performed as previously described [21]. Briefly, the same protein amounts were separated by SDS-polyacrylamide (4–20%) electrophoresis and transferred to a polyvinylidene difluoride (PVDF) membrane (Amersham, Freiburg, Germany). Membranes were subsequently incubated with primary antibodies in a 1:1.000 dilution. Antibodies raised against Aqp2-4 were obtained from Alomone Labs (Jerusalem, Israel) and antibodies directed against phosphorylated Aqp2 at position S256 were a generous gift from M. Knepper (National Institutes of Health, Bethesda, MD, USA). A secondary, horseradish-peroxidase-linked anti-rabbit antibody was used. Membranes were stripped (30 min/60 °C) using stripping buffer (15 g/L glycine, 1 g/L sodium dodecyl sulfate, 10 mL/L Tween 20, pH to 2.2) and reincubated with an antibody directed against glyceraldehyde 3-phosphate dehydrogenase (GAPDH) to ensure equal loading. Quantification of specific signals was performed using the Lumianalyst software (Roche, Basel, Switzerland).

### 2.3. Real-Time PCR

Total RNA and cDNA synthesis were performed as described before [16]. Real-time PCR was performed using the SYBR Green PCR Master Mix with the ABI PRISM 7900 Sequence Detection System. All instruments and reagents were purchased from Applied Biosystems (Darmstadt, Germany). Relative gene expression values were evaluated with the 2^−ΔΔCt^ method using GAPDH as the reference gene [22]. Specific primer pairs were used, and a list with the gene names, accession numbers, and gene symbols is provided in Table 1.

### 2.4. Immunofluorescence

The cellular localization of Aqp2 was determined via immunofluorescence using a commercially available polyclonal Aqp2 antibody (Alomone Labs). First, IMCD cells were fixed in PBS containing 4% paraformaldehyde for 20 min, washed with PBS (3 times, 10 min), permeabilized in PBS containing 0.1% triton X100 (5 min), again washed with PBS (3 times, 10 min), and, to block unspecific binding sites, incubated in blocking solution (0.3% fish skin gelatine in PBS, 20 min, 37 °C). Then, the cover slips were incubated with a specific antibody against Aqp2 (6090–min), washed with PBS (3 times, 10 min), and incubated for 90 min with an Alexa 488-labeled anti-rabbit IgG antibody and DAPI (4′,6-diamidino-2-phenylindole, dichloride; both Molecular Probes, Leiden, The Netherlands). Last, cells were washed with PBS and mounted with Crystalmount (Biomeda, Foster City, CA, USA). Images were taken with a fluorescence microscope (Axiovert Observer Z1, Zeiss, Oberkochen, Germany).

### 2.5. Statistical Analysis

Data were analyzed with one-way ANOVA with a Tukey-post-test or an unpaired *t*-test using GraphPad-Prism 5.0 (San Diego, CA, USA). All values are presented as mean values ± SEM. A *p*-value ≤0.05 was regarded as significant. Asterisks indicate significant changes.

## 3. Results

### 3.1. Effect of LiCl and GSK3 Inhibition on Aqp2 Protein Expression in Primary Cultured IMCD Cells

In the first step, we tested if the primary cultured IMCD cells are a suitable model to analyze the known effect of LiCl on Aqp2 expression. Therefore, IMCD cells were either left untreated or treated for 24 h with 20 mM LiCl. Compared to untreated cells, LiCl induced a massive decrease of Aqp2 protein expression (Figure 1).

GSK3β inhibition by LiCl plays an important role in the development of LiCl-induced NDI [9]. Therefore, we treated the cells in the same way with SB216763, a potent pharmacological inhibitor for GSK3α/β [23], and analyzed Aqp2 expression by Western blot. SB216763 reduced the amount of Aqp2 protein similar to LiCl (Figure 1). These results show that the primary cultured IMCD cells are a suitable model to study the effect of LiCl and GSK3 inhibition on Aqp2 expression.

In the next step, we tested the time- and concentration-dependent effects of LiCl on Aqp2 expression in IMCD cells. Western blot analysis of IMCD cells treated with different LiCl concentrations resulted in a reduction in Aqp2 expression already at 5 mM (Figure 2). We also tested the time dependence of the effect of LiCl treatment on Aqp2 expression in IMCD cells. The results show that at a concentration of 20 mM LiCl, the reduction of Aqp2 expression occurs after 4 h (Figure 2).

We also tested if the SB216763-mediated effect is concentration dependent. Using concentrations between 1 and 20 µM/24 h showed that doses of >10 µM led to a decrease in Aqp2 expression (Figure 3). We also used TWS119, a pharmacological substance described to specifically inhibit GSK3β [15]. Surprisingly, this was followed by a concentration-dependent upregulation in Aqp2 expression (Figure 3).

### 3.2. LiCl and GSK3 Inhibition Have Different Effects on Aqp2 mRNA Expression

To analyze if the downregulation of Aqp2 protein is due to reduced mRNA expression, we measured the amount of Aqp2 mRNA by real-time PCR using the same settings as described above. Treatment of IMCD cells for 24 h with 20 mM LiCl reduced the Aqp2 mRNA expression (Figure 4a). We also observed that LiCl significantly reduced Aqp3 mRNA expression and the same tendency was seen for Aqp4 mRNA and protein expression. The expression of Aqp2 is mediated by the transcription factor cAMP response element-binding protein (CREB) [24], and Aqp2 is also a target gene of tonicity-responsive enhancer binding protein (TonEBP) [25]. Additionally, the aldose reductase (AR) and the betaine transporter 1 (BGT-1) are target genes of TonEBP. Compared to Aqp2, AR and BGT-1 showed significant increases in mRNA expression upon LiCl treatment (Figure 4a).

This indicates that LiCl, at least, does not reduce the action of TonEBP and that the downregulation of Aqp2 on the mRNA level is independent of TonEBP activity.

The use of SB216763 and TWS119 had opposite effects on Aqp2 protein expression. We performed real-time PCR experiments and compared the effects on mRNA expression for selected genes. In contrast to LiCl, SB216763 had no effect on Aqp2 mRNA expression. There was also no effect on AR and BGT-1 mRNA expression. However, it led to decreased expression of the urea transporter type 1 (UTA-1). In line with the observed changes in protein expression, TWS119 treatment was associated with increased Aqp2 mRNA expression (Figure 5).

### 3.3. Downregulation of Aqp2 Is cAMP-Responsive Element-Binding Protein Independent

The application of LiCl together with SB216763 or TWS119 had different effects on Aqp2 protein expression. LiCl together with SB216763 had no additive effect on Aqp2 protein expression. However, using LiCl together with TWS119 reversed the effect of LiCl. No down- or upregulation of Aqp2 expression was observed. Similar results were obtained on the mRNA level. LiCl together with SB216763 had no significant effect on Aqp2 mRNA expression. The treatment of the cells with LiCl and TWS119 reversed the effects compared to the treatment with both substances alone (Figure 6). Since the expression of Aqp2 is also induced by the activity of the cAMP-responsive element-binding protein (Creb), we tested if the treatment had an effect on Creb protein or phosphorylated Creb levels. The treatments had no effect von Creb or phosphorylated Creb protein expression.

### 3.4. Effect on Aqp3 and Aqp4 Protein Expression

Real-time PCR experiments showed that at least LiCl downregulates Aqp3. We performed Western blot analysis using lysates from IMCD cells left untreated, treated with LiCl (20 mM), SB216763 (10 µm), or with TWS119 (10 µm). The results indicated that LiCl also led to downregulation of Aqp3 on the protein level. While SB216763 did not affect Aqp3 protein expression, TWS119 led to increased Aqp3 expression (Figure 7). This effect is similar to that observed for Aqp2. Aqp4 did not show any alterations.

### 3.5. LiCl and GSK3 Inhibition Promotes Lysosomal Degradation of Aqp2

As we did not see a downregulation of the TonEBP target genes AR or BGT-1 on the mRNA level by SB216763, we postulated that the decrease of Aqp2 on the protein level might be induced by changes in protein stability. We therefore tested the effects of lysosome and proteasome inhibitors on LiCl and SB216763-mediated Aqp2 downregulation. We treated the IMCD cells with the proteasome inhibitor MG132 (2 µM) or with the lysosome inhibitor bafilomycin (100 nM) alone or together with LiCl (20 mM) and the GSK3 inhibitor SB216763 (10 µM). MG132 or bafilomycin treatment alone had no significant effect on Aqp2 expression (Figure 8).

The effect of LiCl in the presence of MG132 still led to the downregulation of Aqp2 expression, indicating that LiCl or SB216763 do not promote proteasomal degradation of Aqp2. However, in the presence of bafilomycin, the effects of SB216763 or LiCl on Aqp2 were not evident anymore.

To test whether this effect is also associated with changes on the mRNA level, real-time PCR experiments were performed using the same settings. The results show that the 24-h treatment with MG132 or bafilomycin was followed by a significant reduction of Aqp2 mRNA (Figure 9), unlike the respective effect on protein expression. This treatment was also followed by a significant downregulation of Aqp3, Aqp4, BGT-1, and AR.

Similar results were obtained when chloroquine, another lysosomal inhibitor, was used. Chloroquine alone was associated with an increased Aqp2 protein level. In the presence of chloroquine, the effect of LiCl and SB216763 was abolished (Figure 10).

### 3.6. Bafilomycin Leads to an Accumulation of Aqp2 in Vesicular Structures

It has been shown that there is a continuous shuttling of Aqp2 between the plasma membrane and recycling vesicles [26]. The inhibition of lysosomal degradation by bafilomycin had a beneficial effect on Aqp2 expression in the presence of the LiCl and GSK3 inhibitor (Figure 8). This indicates an accumulation of Aqp2 in the lysosomes. Immunofluorescence analyses showed a high cytoplasmic localization of Aqp2 in untreated cells (Figure 11a). Stimulation of IMCD cells with the lysosomal inhibitor bafilomycin (100 nM for 24 h) led to Aqp2 accumulation in perinuclear vesicles (Figure 11b).

This indicates that Aqp2 is located in bafilomycin-sensitive structures. We also tried to perform colocalization studies using antibodies directed against lysosomal marker proteins (Lamp1 and Rab5); unfortunately, we did not obtain specific signals with the used antibodies.

The localization of Aqp2 depends mainly on its phosphorylation state. Aqp2 phosphorylated at the positions 256 or 269 is accumulated in the plasma membrane [26,27]. We therefore tested if the phosphorylation of Aqp2 prevents accumulation in vesicular structures. We treated the cells with the direct activator of the PKA, dbcAMP (dibutyryl cAMP) [21]. Aqp2 was enriched in the plasma membrane in dbcAMP-stimulated cells (Figure 11d). In the presence of dbcAMP, bafilomycin did not lead to an accumulation of Aqp2 in the vesicular structures and Aqp2 was still localized in the cell membranes (Figure 11e). To test the hypothesis that the phosphorylation of Aqp2 prevents lysosomal accumulation and thereby the GSK3 inhibitor- and LiCl-mediated downregulation, we performed Western blot analysis. We treated the cells with dbcAMP in the presence or absence of the GSK3 inhibitor or LiCl. While LiCl still reduced the amount of Aqp2, the effect of the GSK3 inhibitor was abolished. Similar effects were observed using an antibody directed against the S256-phosphorylated form of Aqp2 (a kind gift from Prof. Mark Knepper [28]; Figure 12).

## 4. Discussion

LiCl as a drug is used in the treatment of bipolar disorders [29]. Compared to other mood-stabilizing drugs, it has a higher efficiency. However, one of the observed side effects of LiCl therapy is NDI, which is associated with a reduction of Aqp2 expression [9]. Several studies have been performed to unravel the cellular and molecular mechanisms involved. One postulated mechanism was related to impaired apical membrane trafficking of Aqp2 expression. LiCl reduces the stimulation of the G_s_ protein, which in turn would inhibit the activity of the adenylate cyclase [30]. This would also reduce the cAMP levels and PKA activity [9]. LiCl reduces the density of V_2_R [31]; therefore, reduced activity of the adenylate cyclase was observed, which has the same effect on Aqp2 trafficking as described above. Moreover, LiCl might cause a defect, which is independent of the classical V_2_R pathway [10,11]. LiCl, for example, inhibits GSK3α/β, which itself has a regulating effect on the Aqp2 water transport [32]. A GSK3β-deficient mouse showed only a mild polyuric phenotype. However, under water deprivation or stimulation with AVP, reduced cAMP levels and in turn a urine concentration disability were observed [14]. In the present study, we used primary cultured IMCD cells to further analyze the underlying mechanisms. We were able to show that LiCl downregulates Aqp2 expression on the mRNA and protein level. This is in line with other studies where LiCl application causes a drastic downregulation of Aqp2 expression, which can only in part be reversed by dDAVP treatment [33] or thirsting [11]. Additionally, in the cortical collecting duct cell line (mpkCCD), a paracrine prostaglandin-mediated effect of LiCl on Aqp2 degradation is observed. Additionally, a prostaglandin-independent effect by LiCl on Aqp2 gene transcription is detected while the action of prostaglandins is coincided to the action of GSK3β [34].

As mentioned above, LiCl has an inhibitory effect on GSK3α/β, which in turn might induce downregulation of Aqp2 [32]. We also observed that LiCl induced downregulation of Aqp3. Using SB216763, a substance that inhibits both GSK3 isoforms with the same potency, we also observed downregulation of Aqp2 on the protein level, while no effect was observed for Aqp3 and Aqp4. The downregulation of Aqp2 is a direct effect, in contrast to the study by Rao et al., where no effect on Aqp2 expression was observed under basal conditions [14]. As they used GSK3β−deficient mice, GSK3α is still present and might have a compensatory effect. This is supported by our results obtained with TWS119, a substance described to inhibit more specifically only the GSK3β isoform [15]. This could also explain why we, compared to the study by Rao et al., observed downregulation already under basal conditions. An unexpected result was that inhibition of GSK3β with TWS119 was associated with increased Aqp2 protein and mRNA expression. In addition, TWS119 increased Aqp3 but not Aqp4 expression. This data implies that the different isoforms of GSK3 have opposite effects on Aqp2 expression and that GSK3α inhibition by LiCl or SB216763 induces downregulation of Aqp2. Additionally, a reduced upregulation of Aqp2 mRNA in GSK3β-deficient mice was only observed under water-deprived conditions, which increases AVP levels in these animals [14]. This is comparable with our results obtained in the IMCD cells cultivated in the presence of dbcAMP. However, in our cell culture model, the stimulation with dbcAMP had a beneficial effect on Aqp2 expression when the cells were treated with the GSK3 inhibitor SB216763. However, dbcAMP was not able to prevent the LiCl-induced downregulation of Aqp2. This is in line with the results obtained by Li et al., where the downregulation of Aqp2 was dissociated from adenylyl cyclase activity [10]. Another study showed that the adenylyl cyclase 6 had only a minor role in LiCl-induced NDI [35].

As described above, Aqp2 mRNA expression is regulated by CREB and TonEBP [25]. To test the cellular mechanisms that lead to downregulation of Aqp2 mRNA induced by LiCl and SB216763, we also examined the expression of proteins and genes that are used as clear target genes of TonEBP, like BGT-1 and AR [25]. In our analysis, BGT-1 and AR mRNA expressions were upregulated by LiCl treatment in contrast to Aqp2 and Aqp3. Since 20 mM LiCl induces a 40 mosmol/kg increase in osmolality, this implies that LiCl might induce TonEBP activity. This indicates that LiCl, at least, does not negatively affect TonEBP activity and we can conclude that the observed changes induced by LiCl treatment are mediated independently of the TonEBP activity. LiCl, however, also led to downregulation of Aqp3 mRNA. Therefore, an additional role of Aqp3 on LiCl-induced NDI cannot be ruled out. This should be considered in order to fully understand the LiCl effect on the collecting duct for the development of NDI.

The persistence of the effect of LiCl on the Aqp2 mRNA level in the presence of the GSK3 inhibitor suggests an enhanced degradation of Aqp2 by LiCl. Other studies have shown that Aqp2 can be degraded in proteasomes [27,36]. In this study, blocking proteasomal degradation via MG132 could not prevent the Aqp2 downregulation after LiCl or after GSK3 inhibition via SB216763. On the other hand, the absence of Aqp2 degradation after inhibition of lysosomal activity by bafilomycin or chloroquine confirmed the key role of the lysosomes in the LiCl- and SB216763-induced Aqp2 degradation, as it was previously shown [34]. Nevertheless, both substances induced a massive downregulation of Aqp2 on the mRNA level. The results obtained here are in line with a recent published study. The authors could show that chloroquine ameliorated LiCl-induced polyuria in mice and that it had a protective effect on the Aqp2 protein level in cortical collecting duct cell lines [37]. For MG132, inhibition of TonEBP could be the reason for the decrease since it is known that proteasome inhibitors like MG132 block the nuclear translocation of TonEBP [38] and its increase in the nucleus [39], which all lead to reduced expression of genes like BGT-1 and the collecting duct aquaporins (Figure 9). The mechanism of how bafilomycin affects the mRNA expression of the analyzed genes remains unclear. The importance of lysosomal degradation of Aqp2 after LiCl is further supported by the immunofluorescence analysis. Bafilomycin-treated IMCD cells showed a perinuclear and vesicular pattern of Aqp2 distribution, in contrast to untreated cells, where Aqp2 was distributed in the cytoplasm. This is in line with previous findings in which bafilomycin-sensitive structures were shown to be involved in Aqp2 recycling [26,40]. The cAMP/PKA-induced translocation of Aqp2 to the plasma membrane could still be observed after lysosomal inhibition by bafilomycin. This reveals the known fact that stimulation of PKA (here via dbcAMP) leads to an insertion of Aqp2 into the plasma membrane in this cell model [41]. This stimulation protects Aqp2 from degradation. Western blot analysis of the protein expression showed a clear effect, indicating that dbcAMP can abolish the SB216763-enhanced degradation process while LiCl still decreases Aqp2 probably due to reduced gene transcription [34] independently of GSK3β.

## 5. Conclusions

In conclusion, using a cell model for lithium-induced NDI, we showed that SB216763 and LiCl were associated with decreased Aqp2 expression. On the one hand, LiCl decreased Aqp2 mRNA, probably facilitated by reduced Aqp2 gene transcription. On the other hand, the amount of Aqp2 protein was also reduced by enhanced lysosomal degradation, which could be prevented by phosphorylation and thereby membrane targeting of Aqp2. The use of TWS119, a substance supposed to inhibit only GSK3β, had the opposite effect. This indicates a major role of GSK3α in LiCl- and SB216763-mediated downregulation of Aqp2.

## Figures and Tables

**Figure 1 cells-09-01060-f001:**
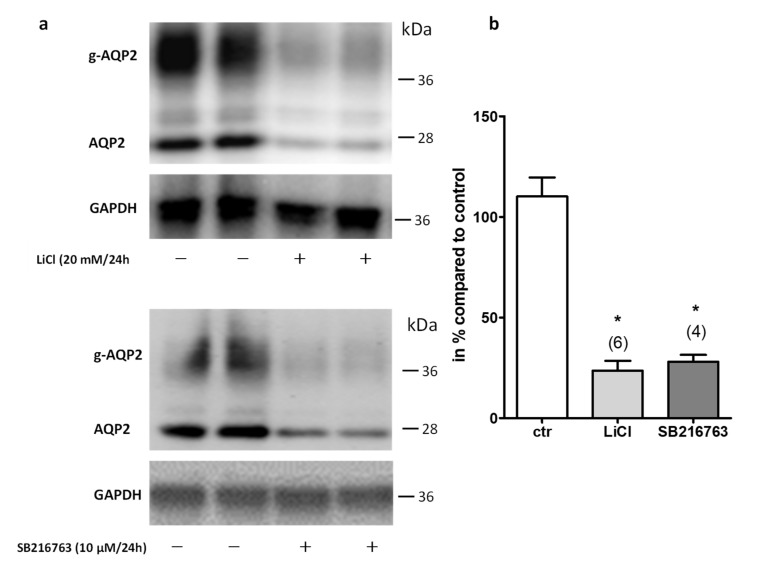
LiCl and GSK3 inhibition reduces Aqp2 expression. The expression of Aqp2 and glycosylated Aqp2 (g-Aqp2) on the protein level in IMCD cells, treated with LiCl (20 mM/24 h) or the GSK3α/β inhibitor SB216763 (10 µM/24 h), was analyzed by Western blot analysis. In the next step, the membrane was stripped and incubated with an antibody directed against GAPDH (**a**). The signal intensities were densitometrically analyzed and the relative changes in Aqp2 expression compared to untreated cells were calculated using GAPDH for normalization (**b**). One-way ANOVA analysis with Tukey post-test was performed and ***** indicates statistically significant differences compared to untreated cells (*p* ≤ 0.05).

**Figure 2 cells-09-01060-f002:**
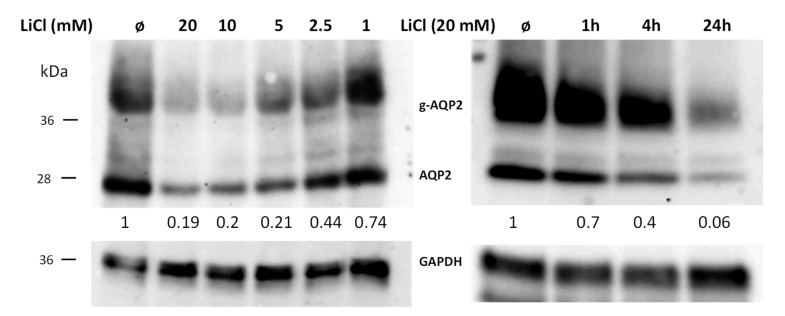
Downregulation of Aqp2 by LiCl is concentration and time dependent. IMCD cells were left untreated and treated for 24 h with different concentrations of LiCl (left panel, concentrations as indicated) or treated for different periods of time (right panel, time as indicated) with 20 mM of LiCl. The expression of Aqp2 was analyzed by Western blot. Afterwards, the antibodies were stripped, and the membrane was incubated with GAPDH. The numbers indicate the relative Aqp2 signal intensities compared to untreated cells (n = 1).

**Figure 3 cells-09-01060-f003:**
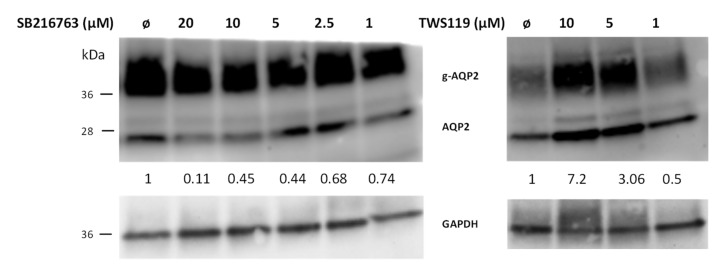
SB216763 and TWS119 have different effects on Aqp2 expression. IMCD cells were left untreated and treated for 24 h with different concentrations of GSK3α/β SB216763 (left panel, concentrations as indicated) or treated for 24 h with different concentrations of GSK3β TWS119 (right panel, concentrations as indicated). The cells were lysed and the expression of AQP2 was analyzed by Western blot. Afterwards the antibodies were stripped, and the membrane was incubated with GAPDH. The numbers indicate the relative Aqp2 signal intensities compared to untreated cells (n = 1).

**Figure 4 cells-09-01060-f004:**
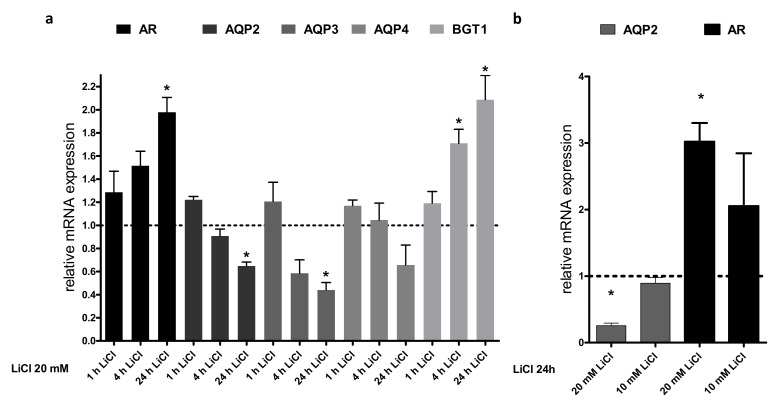
Downregulation of AQP2 mRNA by LiCl is time and concentration dependent. IMCD cells were treated for different time points with LiCl (20 mM). The mRNA expression of AQP2-4, BGT1, and AR was analyzed by real-time PCR and the relative changes compared to untreated cells were calculated (**a**). In the same way, IMCD cells were treated for 24 h with 10 or 20 mM of LiCl and the relative changes in the gene expression of AQP2 and AR were compared to untreated cells (**b**). One-way ANOVA analysis with Tukey post-test, * indicates statistically significant differences to untreated cells (*p* ≤ 0.05); n = 3.

**Figure 5 cells-09-01060-f005:**
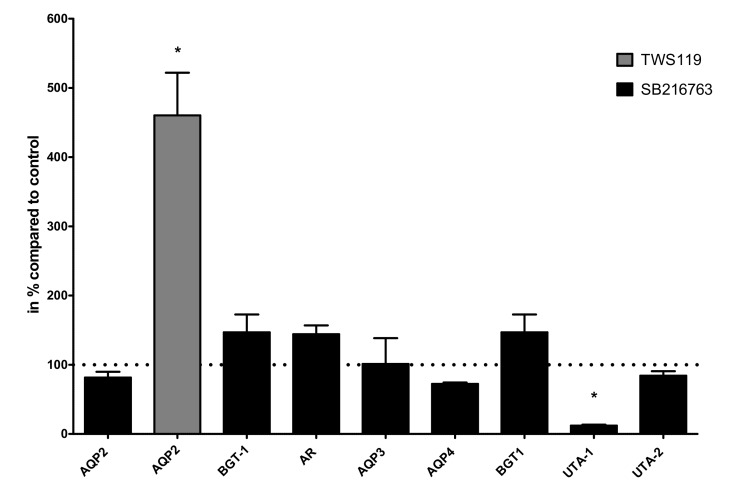
Effect of SB216763 and TWS119 on gene expression. IMCD cells were treated with the GSK3α/β inhibitor SB216763 (10 µM/24 h) and the expression of AQP2, AQP3, AQP4, betaine/GABA transporter 1 (BGT-1), aldose reductase (AR), urea transporter 1 (UTA-1), and urea transporter 2 (UTA-2) was compared to untreated cells using real-time PCR. In the same way as cells treated with the GSK3β inhibitor TWS119 (10 µM/24 h), the expression of AQP2 was compared to untreated cells. One-way ANOVA analysis with Tukey post-test to compare the AQP2 columns, * indicates statistically significant differences compared to untreated cells (*p* ≤ 0.05); n = 3.

**Figure 6 cells-09-01060-f006:**
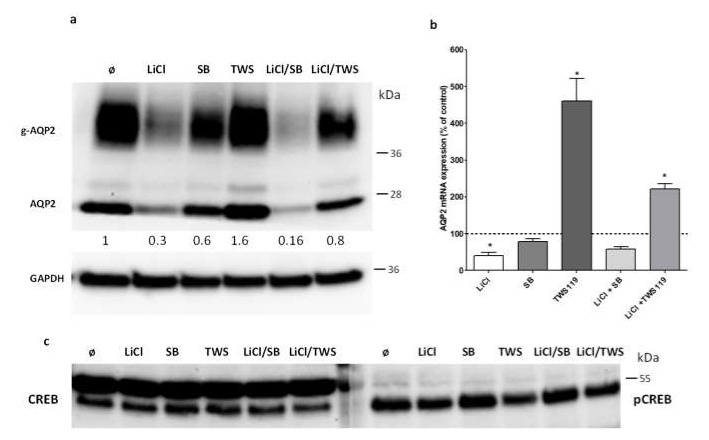
Effect of LiCl treatment in combination with GSK3 inhibition on AQP2 expression. The expression of Aqp2 in IMCD cells treated with LiCl (20 mM/24 h), GSK3α/β inhibitor SB216763 (SB; 10 µM/24 h), GSK3β TWS119 (TWS; 10 µM/24 h), and in combination of LiCl together with SB216763 (LiCl/SB) or LiCl together with TWS119 (LiCl/TWS) was analyzed by Western blot (**a**). Afterwards, the antibodies were stripped, and the membrane was incubated with GAPDH. The numbers indicate the relative Aqp2 signal intensities compared to untreated cells (n = 1). The same setting was used to analyze the Aqp2 mRNA by real-time PCR (**b**). One-way ANOVA analysis with Tukey post-test and * indicates statistically significant differences compared to untreated cells (*p* ≤ 0.05); n = 3. Using the same samples as described in (**a**), the expression level of cAMP-response element-binding protein (CREB, **c**) left panel) and of the phosphorylated form of CREB (pCREB, **c**) right panel) was analyzed by Western blot.

**Figure 7 cells-09-01060-f007:**
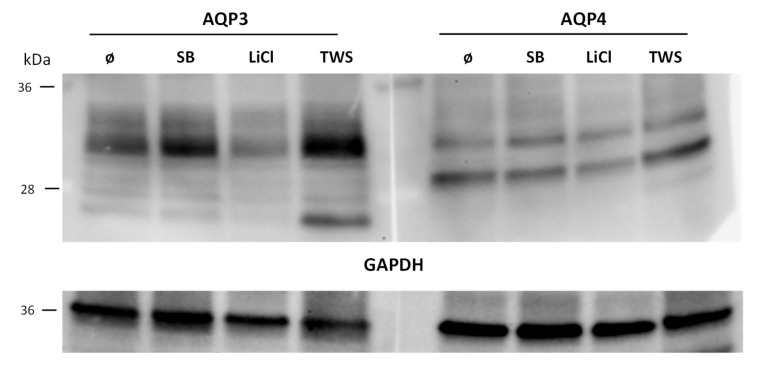
Effect of LiCl and GSK3 inhibition on Aqp3 and Aqp4 expression. The expression of Aqp3 (left panel) and Aqp4 (right panel) in IMCD cells treated with either LiCl (20 mM/24 h), GSK3α/β inhibitor SB216763 (SB; 10 µM/24 h), or GSK3β TWS119 (TWS; 10 µM/24 h) was analyzed by Western blot. Afterwards, the antibodies were stripped, and the membrane was incubated with GAPDH (n = 1).

**Figure 8 cells-09-01060-f008:**
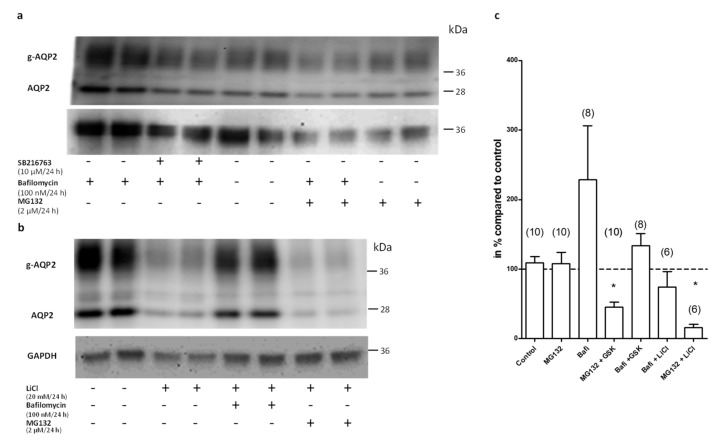
MG132 and bafilomycin alone had no effect on Aqp2 protein expression. The expression of Aqp2 in IMCD cells treated with bafilomycin (Baf., 100 nM/24 h), MG132 (2 µM/24 h), and in combination with SB216763 (GSK, 10 µM/24 h) was analyzed by Western blot (**a**). In the same way, the expression of Aqp2 in IMCD cells treated with bafilomycin (Baf. 100 nM/24 h), MG132 (2 µM/24 h), and in combination with LiCl (20 mM/24 h) was analyzed (**b**). The signal intensities were densitometrically analyzed and the relative changes in Aqp2 expression compared to untreated cells were calculated using GAPDH for normalization (**c**). One-way ANOVA analysis with Tukey post-test; * indicates statistically significant differences to untreated control cells (*p* ≤ 0.05 The number in brackets indicate the numbers of performed experiments.

**Figure 9 cells-09-01060-f009:**
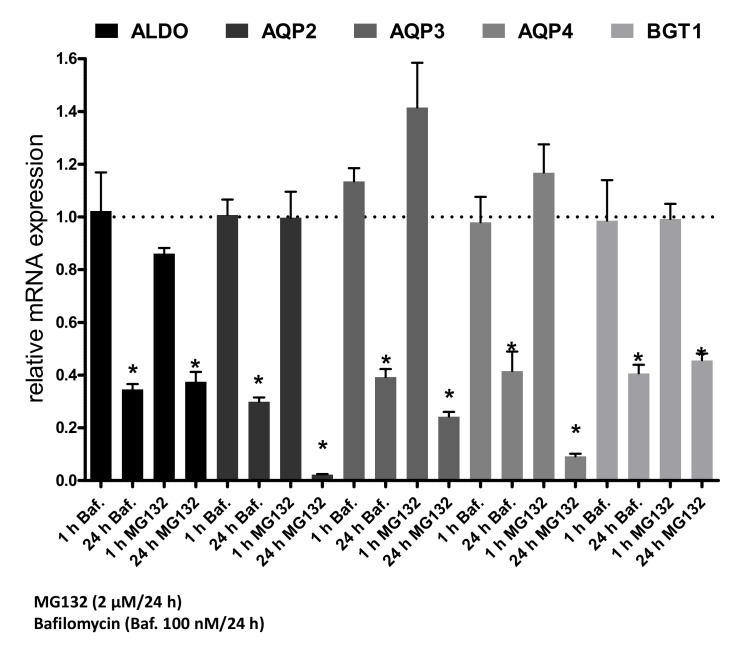
MG132 and bafilomycin induce downregulation of Aqp2 on the mRNA level. IMCD cells were treated for 1 or 24 h with bafilomycin (Baf. 100 nM) or MG132 (2 µM) and the mRNA expression of Aqp2-4, BGT-1, and AR was compared to untreated cells by real-time PCR. One-way ANOVA with tukey post-test was performed and * indicates statistically significant differences compared untreated cells (*p* ≤ 0.05).

**Figure 10 cells-09-01060-f010:**
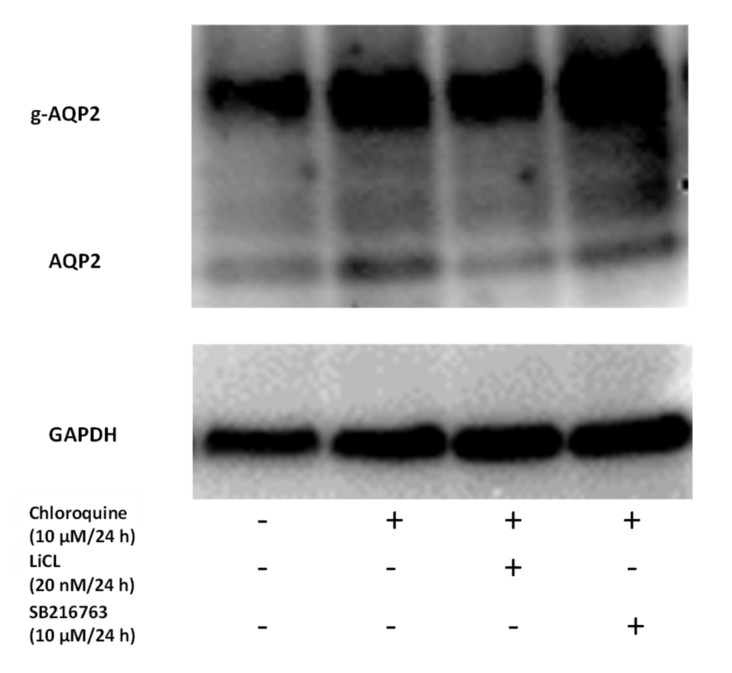
Treatment with chloroquine had a similar effect to bafilomycin. The expression of Aqp2 in IMCD cells treated with chloroquine alone (10 µM/24 h) or in combination with LiCl (20 mM/24 h, or the GSK3α/β inhibitor SB216763 (10 µM/24 h) was analyzed by Western blot. Afterwards, the antibodies were stripped, and the membrane was incubated with GAPDH (n =1).

**Figure 11 cells-09-01060-f011:**
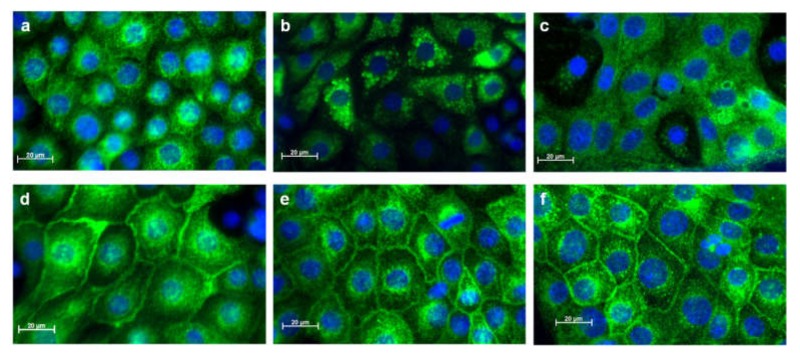
Treatment with dbcAMP prevents the accumulation of Aqp2 in lysosomal structures. IMCD cells were grown on coverslips and were left untreated (**a**), incubated with bafilomycin (100 nM/24 h, **b**), with chloroquine (10 µM/24 h, **c**), incubated in the presence of dbcAMP (500 µM/24 h, **d**), with dbcAMP together with bafilomycin (**e**), or dbcAMP together with chloroquine (**f**). The localization of Aqp2 was analyzed by immunofluorescence using an anti-Aqp2 antibody and an Alexa-488-labeled secondary antibody. The nuclei were stained with diamidino phenylindole (DAPI). Images were taken using a Zeiss Axiovert Observer microscope.

**Figure 12 cells-09-01060-f012:**
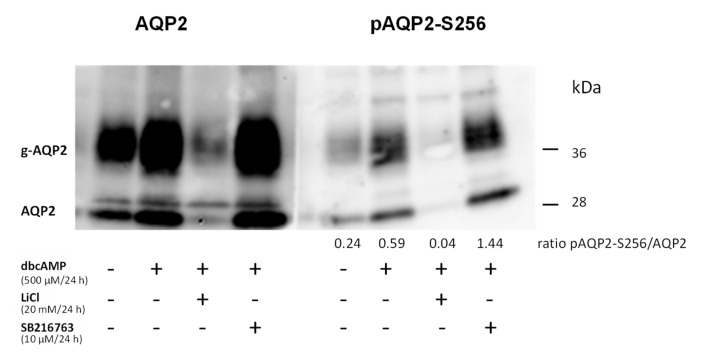
Treatment with dbcAMP prevents downregulation of Aqp2 by GSK3α/β inhibition with SB216763. IMCD cells were left untreated, incubated with dbcAMP (500 µM/24 h) alone or together either with LiCl (20 mM/24 h), or with the GSK3 inhibitor (SB216763, 10 µM/24 h). The expression of Aqp2 or of Aqp2 phosphorylated at position S256 (pAqp2-S256) was analyzed by Western blot analysis. The numbers indicate the relative p256-Aqp2 signal intensities compared to total Aqp2 (n = 1).

**Table 1 cells-09-01060-t001:** Gene names, gene symbols, accession numbers, and primer sequences of genes used for real-time PCR.

Gene Description and Acc. No.	Primer Sequences
Aquaporin-2 (Aqp2)	Sense-TGC ATC TTT GCC TCC ACC GAC GAG Antisense-CAT GGA GCA ACC GGT GAA ATA G
Glycerinaldehyd-3-phosphat dehydrogenase (Gapdh)	Sense-CAT CAA CGA CCC CTT CAT T Antisense-ACT CCA CGA CAT ACT CAG CAC
Aquaporin-3 (Aqp3)	Sense-TTT CAC TGC CCT GGC TGG CTG Antisense-AAG GCG GGG GCT GCT CCA GG
Aquaporin-4 (Aqp4)	Sense-CAT GGG AAA CTG GGA AAA CCA C Antisense-GCG ACG TTT GAG CTC CAC GTC
Betaine/GABA transporter 1 (BGT-1/Slc6a12)	Sense-CCT CCA TGG CCT GTG TAC CGC Antisense-GA GTT CTT GCT TGA CTG GAG AG
Aldose reductase (AR/Akr1b1)	Sense-GTG CTG ATC CGG TTC CCC ATC Antisense-CTC ATC AAG GCG CAC ACC CTC
Urea transporter, member 1 (UTA-1/Slc14a1)	Sense-ATG GCA CTC ACC TGG CAG ACC Antisense-GGC CAA GCA GAA GGA CCA GG
Urea transporter, member 2 (UTA-2/Slc14a2)	Sense-TCT ACG TCA TCA CCT GGC AGA C Antisense-TGC CAG GGT TGT TGG TTG TGA G

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
