# Peer review of "Lithium Chloride and GSK3 Inhibition Reduce Aquaporin-2 Expression in Primary Cultured Inner Medullary Collecting Duct Cells Due to Independent Mechanisms"

_cells, 2020, doi:10.3390/cells9041060_

Round 1
Reviewer 1 Report
Major comments
Figure 1. Data was analyzed with a one-way anova. This test requires that data fulfill the assumption of variance homogeneity. It is unclear if data is normalized to a control value and if all control values therefor equals 1. In that case, the variation of the control is zero and a one-way anova test cannot be used. Please describe in detail the calculations. If the control variance is zero, an appropriate statistical analysis should be used. This applies to other figures as well.
Figure 7, 11 and 13. What was the sample size and how many experiments were performed? All data should be quantified for conclusions to be valid.
Figure 8a. WB images seem to originate from different developments/membranes. In that case, samples should be rerun on the same membrane.
l. 365. It is concluded that TonEBP activity was not affected by the different exposures. But TonEBP activity was never investigated. This conclusion should be weakened/modified in the manuscript.
Minor comments
l. 15: GSK@@ -> correct throughout manuscript.
Molecular weight should be indicated on all WB images.
“LiCL” should be “LiCl”. See e.g. fig. 6 and 7.
Reviewer 2 Report
In this paper, Kaiser and Edemir attempted to clarify the molecular mechanism of LiCl-induced nephrogenic diabetes insipidus in vitro using primary cultured inner medullary collecting duct cells focusing on expression and localization of AQP2 protein. The authors demonstrated that the effects of LiCl, including downregulation of AQP2 transcription and enhancement of lysosomal degradation of AQP2, are independent of GSK3β. This is an interesting study. However, some data obtained by Western blotting were not quantified or appropriately used, and therefore, they were not convincing.
Specific comments:
1. Reduction of AQP2 protein after treating the cells with SB216763 even at a concentration of 20 μM shown in Figures 3 and 6 is not obvious, which looks quite different from that in Figure 1a. As the authors did in Figure 1, quantification is required for these figures.
2. The Western data shown in Figures 8 and 9a are inappropriate since the authors seemed to compare blots on the different membranes. As they showed in the other figures, they should present data on the same membrane. Under this condition, in figure 8, the authors mentioned that they did not see any effect of bafilomycin on the level of AQP2 in IMCD cells. However, according to Figure 8b, expression level of AQP2 in bafilomycin-treated cells seems more than two-fold as compared with untreated cells although the increase in a level of AQP2 as a result of bafilomycin treatment does not seem to be statistically significant probably due to inappropriate use of the data.
3. In figure 13, they attempted to show the effect of AQP2 phosphorylation at Ser256 on the stability of the protein. In this case, the ratio of phosphorylated AQP2 to total AQP2 is important. The authors should quantify the Western blot data and show how much AQP2 was phosphorylated by each treatment.
Minor point
1. In figure 4, they mentioned that LiCl did not affect the action of TonEBP. However, treatment with lithium upregulated transcription of some genes controlled by TonEBP. Thus, the authors cannot rule out the possibility that LiCl affect the action of TonEBP.
2. Page 9, the authors mentioned, “As we did not see a down regulation of the TonEBP target genes AR or BGT-1 on the mRNA level by LiCl and SB216763, we postulated that the decrease of AQP2 on the protein level is mediated by increased degradation of AQP2 induced by GSK3 inhibition.” However, it is unclear why lack of down regulation of the TonEBP target gene transcriptions is related to decreased level of APQ2 as a result of protein degradation.
3. In the discussion section, lines 362-363, the authors mentioned “In our analysis BGT-1 and AR mRNA expressions were up regulated by LiCl and SB216763 treatment in contrast to AQP2 and AQP3.” However, it is not consistent with the result they present in Figure 5, in which no up or down regulation was observed in cells treated with SB216763.
4. The reference 15 and 18 are duplicated.
Some typographical errors:
Sometimes GSK3α and GSK3β are corrupt.
Page 14, line 332, Expression should be expression
Page 15, line 401, be should be BE
Reviewer 3 Report
Dear Authors
This is an interesting study to clarify underlying mechanism of LiCl-induced diabetes insipidus. Although I have no doubt about the quality of the presented work, I strongly recommend to revise Discussion. Discussion is poorly organized, so it is difficult to understand what the authors want to say. For example, subheadings would make it easy for readers to understand. In addition, some schema summarizing this work will help readers to understand the results and discussion.
Specific comments
1) Information of the AQP3 and AQP4 antibodies is required in Materials and Methods.
2) Literature(s) should be cited regarding phosphorylated AQP2 antibody which was given from Knepper.
Thank you very much.
Reviewer 4 Report
Kaiser et.al., were evaluated the effect of Lithium chloride (LiCl) and other glycogen synthase kinase 3 (GSK3) inhibitors primarily on Aquaporin-2 (Aqp2) expression in primary cultured inner medullary collecting duct (IMCD) cells from rats. Authors studied the different concentrations of LiCl and treatment time points on protein and mRNA levels of Aqp2. Further, authors have examined the effects of lysosomal and proteasomal inhibition, cellular localization of Aqp2 in LiCl treated and control cells.
Overall, the authors suggest that the GSK3 involves in reduced Aqp2 expression in LiCl induced nephrogenic diabetes insipidus model IMCD cells. However, specific aims and design of the study is not appropriate enough and several sentence in the manuscript may be right but, not the scientific language. Authors are advised to edit the article professionally before submission. The manuscript in its present state is not acceptable for publication.
Some imperfections listed below:
- Western blot images: Authors are advised to mention molecular markers. See some recently published articles on similar experiments ex: Cells 2019, 8(3), 265; Cells 2020, 9(3), 673
- Subheading 3.4, Line 178: authors states about BGT-1 mRNA expression but the figure 4b is missing the same information.
- Fig 8 and 9: It is suggested to include only LiCl treatment and the loading controls. Both figures may be merged int to one?
- Fig12: Suggested to maintain the same magnification and mention the scale bar. Though the protein expression studies were carried out using immunofluo, manuscript would be strengthened by colocalization studies using membrane and vesicular markers.
Round 2
Reviewer 1 Report
No further comments
Reviewer 2 Report
Regarding Figure 6, the authors should carry out quantitative analysis of Western blotting performed multiple times.
In addition, the numbers written under the images of Western blotting demonstrating dose-dependent effects of LiCl and SB216763 on levels of AQP2 shown in Figures 2 and 3, respectively, are totally the same, which seems strange to me.
Reviewer 4 Report
Authors have addressed all points mentioned in the previous version and improved the article. I have no additional comments.
However, authors are suggested to take great care before publication
Ex: Seems it's a technical error that, the manuscript still showing some blank boxes in place of alpha/beta symbols (in line 240, 318 etc.)
line 219: close the bracket... (20mM/24h
